# Nature vs. Nurture: The Two Opposing Behaviors of Cytotoxic T Lymphocytes in the Tumor Microenvironment

**DOI:** 10.3390/ijms222011221

**Published:** 2021-10-18

**Authors:** Nagaja Capitani, Laura Patrussi, Cosima T. Baldari

**Affiliations:** Department of Life Sciences, University of Siena, 53100 Siena, Italy; baldari@unisi.it

**Keywords:** tumor microenvironment, cytotoxic T cells, exhaustion

## Abstract

Similar to Janus, the two-faced god of Roman mythology, the tumor microenvironment operates two opposing and often conflicting activities, on the one hand fighting against tumor cells, while on the other hand, favoring their proliferation, survival and migration to other sites to establish metastases. In the tumor microenvironment, cytotoxic T cells—the specialized tumor-cell killers—also show this dual nature, operating their tumor-cell directed killing activities until they become exhausted and dysfunctional, a process promoted by cancer cells themselves. Here, we discuss the opposing activities of immune cells populating the tumor microenvironment in both cancer progression and anti-cancer responses, with a focus on cytotoxic T cells and on the molecular mechanisms responsible for the efficient suppression of their killing activities as a paradigm of the power of cancer cells to shape the microenvironment for their own survival and expansion.

## 1. Introduction

The most recent studies on cancer onset and development have pointed the spotlight on the intense crosstalk between cancer cells and a heterogeneous population of other cells that reside in the tumor microenvironment (TME) [1]. Cancer-associated fibroblasts, the most abundant stromal population in the TME, and cancer-associated fibroblasts-derived extracellular matrix factors support the growth and survival of cancer cells by establishing a tumor-promoting niche [2], further assisted by endothelial cells, which also contribute to tumor growth by promoting angiogenesis, invasion, metastasis, and chronic inflammation [3]. Several types of immune cells can also be found in the TME mainly as a result of their active recruitment. Upon identification of tumor cells, immune cells exploit their specific anti-cancer activities to eliminate them. However, notwithstanding their ability to fight tumor cells, immune cells can become pro-tumoral within the TME.

A major step forward in understanding the dual behavior of immune cells in the TME came from the finding that cancer cells themselves implement efficient suppressive strategies toward immune cells in order to escape the tumor-targeted immune responses. The loss of expression of tumor-associated antigens, major histocompatibility complex (MHC) class I molecules and/or co-stimulatory molecules limits the immunogenicity of cancer cells, making them “invisible” to infiltrating immune cells [4]. Additionally, in the TME, tumor cells rewire their metabolism in response to nutritional stress in order to compete for glucose and amino acids, releasing catabolites that become strongly suppressive for immune cells [5]. Immune cells, which should eliminate tumor cells, become, therefore, useless or even harmful. A paradigm of TME-derived immunosuppressive mechanisms is represented by cytotoxic T lymphocytes (CTLs), whose killing functions are inhibited either directly, by suppressing their anti-tumor activity, or indirectly, by the recruitment of immunosuppressive cells and the release of soluble molecules of which cytokines are a major class [6], in the TME [7] (Figure 1).

In the complex TME scenario, immune cells exhibit, therefore, either anti- or pro-tumoral activities, with profound consequences on disease outcome and response to therapy. Of note, immune cell behavior in the TME is only beginning to be understood, as well as being exemplified by CTLs, whose dysfunction in cancer is controlled by molecular mechanisms that have not been fully uncovered [8]. In this review, we will discuss the most recent findings concerning the dual role of immune cells in both cancer progression and anti-cancer responses, focusing then on CTL suppression, one of the most striking features of the tumor cell all-pervasive activity, and on the molecular mechanisms responsible for efficient hampering of their cytotoxic functions.

## 2. The Janus-like Behavior of TME-Populating Innate Immune Cells

Solid tumors and hematologic malignancies have obvious differences in the identity of the cell of origin, which affects the tumor architecture, therefore modulating not only the surrounding stroma, but also the infiltration and effector responses of immune cells. Indeed, while in hematologic malignancies, tumor cells intimately associate with cellular infiltrates of the TME, in solid tumors, the dialogue between neoplastic cells and the TME is codified by the tumor architecture, with tumor cells located in the center, and the surrounding TME making a barrier, which hampers immune cell infiltration, thereby protecting tumor cells from elimination [9].

The extent of immune cell infiltration has become an important evaluation parameter of solid tumors, which, depending on the tumor “temperature”, are classified in “cold” and “hot” tumors. Cold solid tumors, also called non-immune reactive, immune-excluded or immune-desert tumors, produce anti-inflammatory cytokines and have low de novo antigens and few mutations. Hence, these tumors display low or no immune cell infiltrates and easily evade host recognition. On the other hand, high numbers of immune cells infiltrate both the tumor stroma and the tumor tissue itself in hot solid tumors, characterized by an inflammatory state and a high grade of immunogenicity [10]. The multidirectional crosstalk arising in the “hot TME” tunes immune cell transcriptomes and secretomes and rewires their phenotype to either antagonize or promote tumor growth [11]. This is clearly exemplified by *macrophages*, highly represented in the TME, a feature which earned them the name of tumor-associated macrophages (TAMs), that carry out distinct functions depending on signals received from the environment [12] (Figure 1). Chemoattractants, such as CCL2 and CCL5, together with the complement component C5a, are involved in monocyte recruitment to the TME [13] and in the activation of transcriptional programs, which contribute to their functional skewing [14].

According to both the stimuli by which their polarization is induced and the type of cytokines and transcription factors produced, TAMs are classified as M1 or M2 [15] (Figure 1). M1-polarized macrophages, usually considered tumor-killing cells, produce pro-inflammatory cytokines and reactive oxygen and nitrogen species to exert anti-tumor and immune promoting activities [16]. However, interleukin (IL)-4 and IL-13 secreted by T helper 2 (Th2) cells, eosinophils and basophils elicit an alternative polarization of TAMs to M2 macrophages, which in turn promote vascularization, tumor growth and invasiveness, cancer cell survival, and immunosuppression, all resulting in tumor progression [16]. In healthy tissue, macrophages exist in both M1 and M2 phenotypes. However, in progressive cancers, their balance shifts toward the M2 phenotype, with M1 macrophages mainly populating regressing tumors [17]. Interestingly, chemokines, colony-stimulating factors, and TGF-β secreted by tumor cells, together with other soluble factors provided by immune and stromal cells (interleukins, immune complexes), promote and sustain macrophage skewing to the M2 cancer-promoting phenotype [18].

*Neutrophils*, which represent the traditional first line of defense against infection [19], are also found to be associated to many types of tumors (tumor-associated neutrophils, TANs). TANs contribute to tumor clearance by releasing cytotoxic compounds contained in their granules to destroy malignant cells [20] and by secreting cytokines and chemokines to recruit other immune cells with anti-tumor activity [21]. However, in aggressive neoplasias, TANs have been found to sustain tumor progression by acquiring a pro-tumorigenic profile [22] with high expression of tumor growth-promoting factors [23]. As observed for macrophages, TANs are classified in the N1 and N2 phenotypes, both derived from the same initial population that polarizes under the influence of external stimuli [23] (Figure 1).

Another main type of tumor-promoting immune cells within the TME is represented by *myeloid-derived suppressor cells* (MDSCs), classified as polymorphonuclear (PMN)-MDSC or monocytic (M)-MDSC, reflecting their similarities to neutrophils and monocytes, respectively [24] (Figure 1). The crosstalk between MDSCs and cancer cells is critical for tumor development [25]. MDSCs are recruited to and proliferate in the TME in response to the cytokines and chemokines present in the tumoral milieu [26]. Not surprisingly, the extent of infiltration of these cells within tumor tissues is associated with poor prognosis [27]. Once infiltrated in the TME, MDSCs support tumor growth, on the one hand, by enhancing angiogenesis and promoting metastasis [28] and, on the other, by inhibiting T cell functions through the production of immunosuppressive factors [25].

While not abundantly represented, *dendritic cells* (DCs) are a key component of the TME [29] (Figure 1). As professional antigen presenting cells (APCs), DCs recognize dangerous cells and migrate to the draining lymph node, where they provide the co-stimulatory signals for anti-tumor CD8^+^ T-cell priming [30,31]. In line with the dual role of immune cells in the TME, DCs can also acquire an immunosuppressive phenotype that results in immune tolerance and tumor dissemination [32]. Factors released in the TME, such as vascular endothelial growth factor (VEGF) or tumor-derived mediators, can impair the antigen-presenting ability of DCs, eventually suppressing their anti-tumoral activities [33,34]. Furthermore, under the hypoxic conditions found in TME, DCs express receptors usually found on myeloid cells to trigger pro-inflammatory signals [35].

*Natural killer* (NK) cells are a heterogeneous population of innate immune cells with inherent capabilities in both recognizing and killing cancer cells. The presence of NK cells in the TME correlates with disease outcome in a variety of cancers, emphasizing the critical role that NK cells play in anti-tumor immune responses [36] (Figure 1). However, as for the other cell types described above, various alterations were recently found in the NK cell phenotype, which alter their functions and contribute to immune evasion in cancer patients. While on the one hand, NKs kill malignant cells expressing ligands for NK-specific surface receptors, such as the natural killer group 2D (NKG2D) ligand MIC-A [37], on the other hand, they paradoxically select and promote the expansion of neoplastic clones that develop mutations, which reduce the expression of NK-receptor ligands, making them resistant to immune attack [36].

## 3. Adaptive Immune Cells

Adaptive immune cells are considered the most specific and potent weapons against foreign and dangerous molecules. However, notwithstanding their antigen selectivity, in specific settings, they can become dysfunctional or even extremely dangerous. This is clearly exemplified by autoimmune diseases, exacerbated cytokine storms as observed in COVID-19 patients [38], and cancer, where adaptive immune cells do not function correctly, thereby favoring the onset and development of pathologic conditions. Lymphocytes recruited to the TME, referred to as tumor-infiltrating lymphocytes (TILs), are a heterogeneous population of adaptive immune cells that include the Th1, Th2, Th17 and the recently identified Th9 subsets, regulatory T and B cells (Tregs and Bregs), CTLs and B lymphocytes [7,39,40,41].

Each *Th subset* plays a distinct role in cancer development. While the IL-2- and interferon gamma (IFNγ)-producing Th1 subset has been shown to play an essential role in the induction and persistence of antigen-specific CTLs, acting therefore as an anti-tumoral Th subset, the Th2 and Th17 subsets act as pro-tumoral subsets in a cytokine-dependent manner [39,42]. Of note, naïve CD4^+^ T lymphocytes undergo polarization to Th subsets in response to a specific cytokine milieu, which in the TME is composed of a mixture of soluble factors belonging to pro- and anti-tumoral classes, with one class overcoming the other depending on tumor type and prognostic status. Hence, the balance among pro- and anti-tumoral Th subsets is regulated by the TME, with tumor cells themselves exerting a skewing activity toward pro-tumoral and immunosuppressive Th phenotypes [39].

A frank tumor-promoting activity is exerted by *Tregs*, CD4^+^ T lymphocytes expressing CD25 (the α subunit of IL-2 receptor) and the transcription factor Foxp3. Tregs accumulate in a chemokine-dependent manner in tumor sites, especially those harboring large immune cell infiltrates, where they exert potent suppressive activity not only toward other T cell subsets, but also toward B cells, NK cells, DCs and macrophages via humoral and cell–cell contact mechanisms [43,44]. The anti-tumor activity of Tregs is witnessed by the fact that their presence in the TME is associated to unfavorable prognosis and reduced overall survival [45]. Tregs are recruited to the TME by chemokines secreted by tumor cells. Here, they prevent the anti-cancer response of effector T cells through multiple mechanisms that include (i) depleting IL-2 from their surroundings through their high affinity IL-2 receptor, making this cytokine unavailable to other effector T cells; (ii) constitutively expressing the checkpoint protein CTLA-4, which binds to CD80 and CD86 on APCs, thereby impairing their co-stimulatory activity toward effector T cells; (iii) secreting cytokines, such as IL-10, IL-35, and TGF-β, which suppress the activity of both APCs and effector T cells, and releasing lytic granules that directly kill these cells; and (iv) producing adenosine via the nucleotidase activity of CD39 and CD73, which provides immunosuppressive signals to both effector T cells and APCs through engagement of the adenosine receptor A2AR [43,44].

*Tumor infiltrating B lymphocytes* (TIL-B) have been found in the TME of several cancer types, among which include breast cancer [46], melanoma [47], and non-small-cell lung carcinoma [48,49]. Similar to innate immune cells, TIL-B cells play a controversial role. On the one hand they serve as potent APCs to activate T cells and promote anti-tumor immunity, as witnessed by the reported association of B cell infiltration with favorable tumor prognosis [50]. On the other hand, TIL-B cells harbor tumorigenic activities. Immune complexes, formed by antigens bound to antibodies secreted by infiltrating conventional B cells in the tumor milieu, engage Fc receptors on myeloid cells, inducing chronic inflammation-dependent tumor growth [51]. Furthermore, TIL-B cells produce lymphotoxin and vascular endothelial growth factor (VEGF) at the tumor site, which promote angiogenesis and support tumor progression [52,53]. The tumor-promoting ability of B cells is mainly mediated by a subgroup of B cells known as regulatory B cells (Bregs) [54]. By secreting the suppressive cytokines IL-10, IL-35 and TGF-β, Bregs suppress CD4^+^ T cell proliferation and promote Foxp3 expression in Tregs [40,55], thereby favoring immunosuppression and tumor development.

### 3.1. Cytotoxic T Cells (CTLs)

CTLs, the main subset of lymphocytes with cytotoxic activity toward cancer cells, are professional effector T cells that develop from activated naïve CD8^+^ T cells. Two types of stimuli are required to elicit differentiation of CD8^+^ T cells to CTLs following antigen recognition: a first priming signal triggered by interaction of CD70 and B7.1/.2 (CD80/CD86) on DCs with the respective receptors CD27 and CD28 on CD8^+^ T cells, and a second help signal provided by CD4^+^ T cells via CD40-CD40L interaction [8,56]. Specific anti-cancer activity of CTLs has been proven for several tumor types, such as melanoma [57], breast cancer [58], lung cancer [59], hepatocellular carcinoma [60], glioblastoma [61] acute and chronic leukemias [62,63], lymphomas [64], and histiocytoma [65]. High frequencies of tumor antigen-specific CTLs have been related to anti-tumor immune responses and favorable disease outcome [66,67].

The classical picture of T cell–mediated cytotoxicity is based on the formation of a polarized structure between CTLs and target cells, known as the immune synapse (IS). During IS formation, the reorganization of receptors and molecules that are involved in recognition and adhesion leads to the formation of specialized functional domains at the interface between the CTL and target cell. The mature IS consists of three concentric regions: the central supramolecular activation cluster (cSMAC), characterized by the presence of T cell receptors (TCRs) and associated signaling molecules; the peripheral SMAC (pSMAC), enriched in LFA-1 and other adhesion molecules; and the distal SMAC (dSMAC), where receptors with bulky ectodomains are excluded, and with an underlying dense ring of filamentous actin (F-actin) [68]. Active TCR signaling is accompanied by the centripetal movement of TCR microclusters from the periphery to the cSMAC, where they are internalized and either recycled or delivered for degradation, or alternatively released as ectosomes [69,70]. Together with the TCRs, other co-stimulatory or inhibitory molecules can be delivered to the IS either by lateral mobility along the plasma membrane or through polarized vesicular trafficking [71]. A prerequisite for IS assembly and function is the acquisition of cell polarity marked by the translocation of the microtubule organizing center (MTOC) toward the synaptic interface [72,73], a complex event coordinated by the cytoskeleton along with motor proteins [74]. CTL polarity allows for the directional release of their killing machinery onto the target cells, leading to their apoptotic demise.

CTLs exert their tumor-specific killing activity mainly through the release of cytotoxic granules (CGs). CG-mediated cytotoxicity is triggered by TCR engagement by MHC class I-associated peptide antigen on the target cell, which promotes the polarized secretion of CGs in the synaptic cleft. This process involves the association of CGs with the microtubules and their dynein-mediated, minus-end directed transport toward the centrosome, which is in close apposition with the synaptic membrane [75]. There, CGs dock and release their contents in a process dependent on Ca^2+^ and SNAREs [76]. Morphologically, CGs are characterized by a distinctive dense core, containing a number of cytotoxic components, including the pore-forming protein perforin (Prf1) and a battery of proteases known as granzymes (Gzm), which are packed together on the anionic proteoglycan, serglycin (Srgn). Although alternative models of CG-mediated killing have been proposed, the most established model posits that Prf1 polymerizes on the target cell membrane to form pores that allow for the entry of the Gzms, which cleave critical intracellular substrates controlling cell death and survival [77,78,79]. In addition to these cytolytic components, CGs also contain lysosomal hydrolases, such as cathepsins and β-hexosaminidase, and lysosomal membrane proteins, such as CD63, LAMP1 and LAMP2 [80], which highlights their lysosomal origin. This is further supported by the fact that Gzms are transported to CGs via the CI-mannose 6-phosphate receptor (MPR) [81], which is exploited for the transport of acid hydrolases to lysosomes.

In addition to CG secretion, CTLs exert their killing activity through the activation of the Fas apoptosis pathway in target cells. CTLs have an intracellular store of FasL associated with secretory lysosomes that have been identified as multivesicular bodies [82]. FasL is sorted to the secretory lysosomes by a mechanism involving a proline-rich domain in its cytoplasmic tail [83] as well as FasL phosphorylation and ubiquitylation [82]. Following TCR engagement, FasL-enriched vesicles are released at IS, where they bind Fas on Fas-bearing target cells, triggering a signaling cascade that leads to the activation of caspases and target cell death [84]. FasL activity is tightly regulated both transcriptionally [85] and post-transcriptionally [86]. Although both FasL “granules” and CGs are lysosome-like organelles, the different protein compositions, kinetics of release and responsiveness to TCR strength indicate that they may represent two different classes of cytotoxic organelles that cooperate to allow for serial target cell killing by CTLs [87].

Recently, using supported lipid bilayers (SLBs) functionalized with anti-CD3 Fab and LFA-1 as a surrogate APC to promote IS formation [88], Balint and colleagues identified new cytotoxic multiprotein complexes released by CTLs, which they referred to as supramolecular attack particles (SMAPs) [89]. Through a mass spectrometry analysis of the material captured by SLBs after CTL removal, they found that SMAPs have a cytotoxic core of Prf1, GzmB and Srgn, surrounded by a shell of glycoproteins, of which thrombospondin-1 (TSP1) and galectin-1 (Gal-1) are prominent components. They showed that within CTLs, SMAPs are stored in multicore granules and that, following release, they can kill cells autonomously [89] (Figure 2).

Similar to other secretory lysosomes, CG mobilization and secretion first requires activation through the TCR complex. It is well established that the strength of TCR signal affects polarized CTL secretion. TCR engagement by MHC-I-bound cognate ligand results in the activation of the Src family tyrosine kinases, Lck and Fyn [90], which allows for recruitment of the tyrosine kinase ZAP-70 by phosphorylating the ITAMs of the CD3 complex subunits [91]. Downstream ZAP-70 activation, adaptor molecules among which the linker for activation of T cells (LAT) and the SH2 domain-containing leukocyte protein of 76 kDa (SLP-76) are recruited to the nascent lytic synapse to help mobilize several key signaling modules, leading to MTOC polarization, Ca^2+^ signaling and cytoskeletal reorganization [90]. Other pathways that are activated following TCR engagement are involved in selective cytotoxic granule movement toward the IS, including the kinase PKCδ [92], the phospholipase Cγ1 (PLCγ1)- and diacylglycerol kinase α-dependent synaptic accumulation of diacylglycerol [93], and the fine tuning of the local concentration of specific phosphoinositides by lipid kinases and phosphatases [94]. Molecules implicated in other cellular functions, such as the cell cycle-related serine/threonine kinase Aurora A kinase [95], the ciliary protein Bardet Biedl syndrome 1 [96] and the ciliogenesis pathways orchestrated by Hedgehog signaling [31], were recently implicated in lytic synapse formation and CTL-mediated killing, only to mention a few.

### 3.2. Altered Killing Capacities of CTLs in Cancer

Although antigen-driven TCR activation in the presence of co-stimulatory signals leads to the generation of CTLs able to effectively kill their specific cell target, in cancer patients, the persistence of tumor-derived antigens gradually dampens CTL functions. Both hot solid tumors and hematologic malignancies show a profound subversion of the complex molecular machinery exploited by CTLs to kill tumoral target cells, a process known as T-cell exhaustion [97]. This functional state, caused by the continuous antigen-driven activation of CD8^+^ T cells, leads to the upregulation of receptors with inhibitory function, known as immune checkpoints, and to the subsequent subversion of the tight balance between co-stimulatory and inhibitory molecules that controls both the duration and the outcome of the signaling cascade initiated by the TCR (Figure 2). Natural consequences of this imbalance are dampened TCR-dependent responses. These include impaired activation of key TCR-dependent signaling molecules [98,99,100], abnormalities in IS architecture [99,101,102], and a dysfunctional lytic machinery, with decreased expression of Gzms [103] and impaired cytotoxicity [63]. Furthermore, exhausted T cells in the TME lose their proliferative potential and their ability to produce cytokines, such as IL-2, tumor necrosis factor-alpha (TNF-α), and IFNγ [97]. Tumor-specific CTLs display hallmarks of T cell exhaustion and dysfunction in several types of human cancers, including, among others, melanoma [104], ovarian cancer [105], non-small cell lung carcinoma (NSCLC) [106], Hodgkin lymphoma [107], and chronic lymphocytic leukemia [63,97,108].

A paradigm of this inhibitory signaling module is the surface co-inhibitory receptor cytotoxic T-lymphocyte-associated protein 4 (CTLA-4), also known as cluster of differentiation 152 (CD152), which consists of two isoforms, a membrane-bound receptor isoform (mCTLA-4) with both extracellular and intracellular domains, and a soluble isoform (sCTLA-4) with only the extracellular domain for ligand-binding [109]. CTLA-4 binds the co-stimulatory molecules B7.1 and B7.2 expressed on the surface of APCs [110] with an approximately 10–20-fold higher affinity than the surface co-stimulatory receptor CD28, thereby competitively inhibiting CD28 binding to B7. Additionally, the CTLA-4-mediated trans-endocytosis of B7s on neighboring cells results in surface B7 depletion, which contributes to the suppression of CD28 co-stimulation [111]. The intracellular domain of CTLA-4 has also been implicated in the inhibition of T cell signaling. While B7-engaged CD28 delivers a phosphoinositide 3-kinase (PI3K)-dependent co-stimulatory signal for T cell activation, CTLA-4 triggers an inhibitory signal [112,113], which hampers TCR-mediated activation of signaling molecules [114]. Evidence suggests that this inhibitory activity relies on the ability of CTLA-4 to associate with the serine/threonine phosphatase PP2A [115] and with Src homology 2 (SH2) domain-containing phosphatase (SHP)-1, which counteract the phosphorylation steps that are critical for T cell activation [116]. PP2A dephosphorylates and inhibits PI3K, directly antagonizing CD28 signaling, while SHP-2 both represses TCR phosphorylation and stimulates ERK activation [117,118]. CTLA-4 engagement also interferes with CD28 localization at the cSMAC [119] and leads to the disruption of TCR microcluster formation [120], impaired IS assembly and T cell anergy [121] (Figure 3A). Particularly affected by CTLA-4 engagement is the activity of several transcription factors, including nuclear factor-κB (NF-κB), AP-1, and nuclear factor of activated cells (NF-AT) [122], cytokine production, cell cycle, which is usually arrested at G1 [123,124], and glycolysis [125].

The TME indirectly deprives CTLs of the metabolic nutrients required for their survival and activities, and is exemplified by the accumulation of the ion K^+^ in the interstitial fluid of the TME, which suppresses the activity of amino acid and glucose transporters, thereby contributing to starve T cells [126]. Interestingly, CTLA-4 expression also affects T cell metabolism by promoting downregulation of the glutamine transporters SNAT1, SNAT2 and the main glucose transporter Glut1, ultimately preserving the metabolic profile of unstimulated T cells in the TME and further contributing to suppress T cell activities [5,127].

CTLA-4 was found to be expressed on CTLs isolated from several tumor types (reviewed in [128]), where it contributes to suppress host immune surveillance. CRISPR-mediated knock out of CTLA-4 has indeed been found to enhance the anti-tumor activity of CTLs [129,130]. Hampered antigen-driven signaling, together with downregulation of the production of IFN-γ and of glutaminase, which promotes and sustains T cell metabolism in the glucose-deprived TME [131], impairs the ability of cytotoxic T cells to fight tumors. Furthermore, in contrast to CD28, which is constitutively expressed at the T cell surface, CTLA-4 expression is induced both via de novo transcription and via trafficking from intracellular compartments, where it is sequestered in naïve T cells, to the cell surface, with maximal expression occurring two to three days following T-cell activation [4].

Along with CTLA-4, the transmembrane type I molecule lymphocyte-activation gene 3 (LAG-3; also known as CD223) was found to contribute to immune escape in cancer. Its peculiar structure, composed of four extracellular immunoglobulin domains, the first of which, containing an extra proline-rich loop with high binding affinity for MHC class II molecules, mediates its association with the TCR/CD3 complex, making it remarkably similar to the co-receptor CD4. However, as opposed to CD4, the intracellular region of LAG-3 inhibits signaling downstream of the TCR [132], resulting in decreased T-cell proliferation and cytokine production [133] and contributing to the onset of exhausted phenotypes (Figure 3B). Although the molecular mechanism underlying the immunosuppressive function of LAG-3 remains as yet unknown, the general agreement is that discrete motifs, which are conserved in other mammals and contain a potential phosphorylation site at position S454, act, following MHC class II binding, by recruiting or excluding signaling mediators to or from the IS [132]. In the early 2000s, a conserved ‘KIEELE’ motif containing a single lysine residue (K468) was found, whose mutation abrogated the inhibitory activity of LAG-3 [134]. However, these data have not been confirmed and the molecular mechanism underlying the inhibitory activity of LAG-3 remains to be defined. LAG-3 ligands other than the MHC class II were recently identified, which might contribute to immune regulation by triggering or blocking signaling cascades, including fibrinogen-like protein 1, whose upregulated expression correlates with the development of solid tumors [135] and Galectin-3, a 31-kDa lectin that suppresses T cell effector functions via LAG-3 in mice [136]. Similar to CTLA-4, LAG-3 is largely retained in early and recycling endosomal compartments, and rapidly translocates to the plasma membrane following T cell activation [137], suggesting that its subcellular localization might concur to immune suppression.

In CTLs, LAG-3 negatively regulates proliferation and homoeostasis and promotes exhaustion [138,139]. Notably, it was found to be highly expressed in several tumor types [140], where it correlates with marked dysfunction of CD8^+^ TILs [141], aggressive phenotypes and overall poor prognosis [132]. Of note, in some cancer types, including breast cancer, gastric cancer and esophageal squamous cell carcinoma, LAG-3 plays anti-tumoral functions [142,143,144], underscoring the importance of a complete understanding the full range of biological functions of LAG-3 in different tumor contexts for translation to the clinics.

Discovered in the early 1990s as a transmembrane protein involved in T cell apoptosis, programmed death-1 (PD-1) is a co-inhibitory checkpoint and a marker of T cell exhaustion. The high expression in neoplastic cells from several cancer types of its ligands, programmed death-ligands 1 and 2 (PD-L1 and PD-L2), surface molecules involved in the suppression of T cell responses in vivo [145] have drawn, on the PD-1/PD-L1 axis, the attention of the scientific and pharmaceutical community [9,146]. PD-1 is a 288-aa protein consisting of an N-terminal immunoglobulin domain, a transmembrane domain, and a cytosolic tail containing two motifs essential for its inhibitory functions, named immunoreceptor tyrosine-based inhibitory motif (ITIM) motif and immunoreceptor tyrosine-based switch motif (ITSM), respectively [146]. Following antigen recognition, PD-1 binding to its ligands, PD-L1 [147] and PD-L2 [148] expressed on tumor cells and APCs, respectively, elicits tyrosine phosphorylation of the cytoplasmic ITIM and ITSM motifs [149]. Interestingly, while mutation of the ITIM motif has little effect on either the signaling or functional activity of PD-1, mutation of the ITSM motif abrogates the ability of PD-1 to dampen cytokine production and T cell expansion [150]. Transiently phosphorylated ITSM recruits the tyrosine phosphatase SHP-2 [149]. In turn, SHP-2 (i) triggers a positive feedback loop by linking two PD-1 molecules together to form active PD-1 dimers [151], and (ii) dephosphorylates tyrosines within proteins critical for TCR signaling, such as CD3ζ, ZAP-70 and PKCθ [152], thereby downregulating T-cell activation signals. CD28 is also dephosphorylated by PD-1 [153]. However, the overall PD-1 inhibition of T cell responses was found to be comparable in the presence or absence of CD28 co-stimulation, suggesting that CD28 dephosphorylation is not required for the inhibitory activity of PD-1 [154] (Figure 3C).

Although considerable progress has been made in our understanding of the biology of PD-1, its underpinning suppressive mechanism remains, in part, unclear. The ability of PD-1 to block T cell activation following antigen recognition seems to require its recruitment to co-stimulatory microclusters in close proximity to the IS to become rapidly phosphorylated by Src family kinases [150,155]. Within these negative co-stimulatory microclusters, proximal TCR signaling molecules become dephosphorylated [149]. Furthermore, engaged PD-1 has been found to invade the CD2 “corolla”, a membrane region localized at the outer edge of the mature IS which contains engaged CD28, ICOS, and other co-stimulatory molecules, thereby suppressing CD2-mediated amplification of TCR signaling [101]. Of note, a recent work by Tocheva and colleagues [156] identified a complex and branched PD-1-regulated dephosphorylation network, which extends far beyond the expected proximal TCR signaling and whose functional consequences involve cellular scale events, including reduced or suppressed cytoskeletal reorganization and IS maturation [156]. The PD-1-triggered signaling cascade also inhibits the PI3K/Akt/mTOR axis, essential to upregulate glucose metabolism in effector T cells, suggesting a role for PD-1 in modulating CTL metabolism [157].

PD-1 is expressed by a variety of innate and adaptive immune cells, including NK cells, monocytes, DCs, NKT cells, T cells, and B cells [158]. Its expression is high in CD8^+^ TILs and increases exponentially along with tumor growth [159], strongly underscoring its role as promoter of tumor aggressiveness. In turn, as is extensively documented, tumor cells upregulate surface PD-L1 and PD-L2 [160,161] that engage PD-1 on CTLs, thereby triggering the immunosuppressive signaling cascade described above.

Of note, PD-1 expression impairs CTL effector function by downregulating glycolysis, increasing the rate of fatty acid β-oxidation and markedly decreasing mitochondrial respiration, thereby supporting CTL persistence in the tumor but preventing their cytotoxic activities [127,162]. Furthermore, elevated Akt signaling in TILs potently represses the transcriptional co-activator PGC-1α, a key regulator of mitochondrial biogenesis [162]. Altogether, these data provide evidence that the suppressive mechanism exploited by PD-1 in the TME is in part related to its effects on the metabolic rewiring of CTLs. On the other hand, overexpression of PD-L1 enhances glucose uptake on tumor cells, further depriving T cells use of this critical energy substrate [163].

To the group of immunosuppressor molecules also belongs T cell immunoglobulin and mucin domain-containing protein 3 (TIM-3), a member of the TIM family of immunoregulatory proteins originally identified as a T cell-specific molecule and now known to be expressed by other immune cells, including Tregs, myeloid cells, NK cells and mast cells [164]. Its heterogeneous structural organization—an N-terminal immunoglobulin domain with five noncanonical cysteines, a mucin stalk, a transmembrane domain and a cytoplasmic tail—does not contain any obvious inhibitory signaling motif. However, its cytoplasmic tail is characterized by five conserved tyrosines, of which three are of unknown function, while the other two, Y256 and Y263, mediate its mutually exclusive interaction with HLA-B-associated transcript 3 (BAT3) [165] and with the tyrosine kinase Fyn [164,166]. TIM-3 localizes in membrane lipid rafts and becomes recruited to the IS following T cell activation [167], where it can interact with both BAT3 and the tyrosine kinase Lck [168]. The current hypothesis is that TIM-3, in its unbound state, promotes T cell activation by binding BAT3 and recruiting active Lck, thereby enhancing TCR-proximal signaling [167]. Both soluble TIM-3 ligands, the lectin Galectin-9 and the adhesion molecule carcinoembryonic antigen-related cell adhesion molecule 1 (CEACAM1) trigger Y256 and Y263 phosphorylation [169]. Upon phosphorylation, BAT3 is released from TIM-3, thereby shifting the function of TIM-3 from activation to suppression. TIM-3 can indeed now bind Fyn, which, by recruiting the Lck inhibitory kinase Csk, can in turn suppress antigen-dependent signaling. Moreover, in CTLs, TIM-3 colocalizes with the receptor phosphatases CD45 and CD148, an interaction that is enhanced in the presence of Galectin-9 [168], further highlighting the suppressive activity of TIM-3 [164] (Figure 3D).

TIM-3 expression is increased in CD8^+^ TILs in solid tumors, and correlates with poor outcome in several tumor types, among which include neck squamous cell carcinoma [170], urothelial carcinoma [171] and colorectal cancer [172]. However, much remains to be understood about the circuitry by which TIM-3 operates to mediate its effects in different tumoral contexts.

Other mechanisms that promote CTL dysfunction but do not rely on the expression of inhibitory receptors on CTLs or the respective ligands on tumor target cells have been also discovered. Among these, it is worth mentioning the Ras GTPase-activating protein (GAP) Ras protein activator-like 1 protein (Rasal1), which inhibits Ras/MAPK activation and whose reduced expression in cancer cells is linked to tumor progression [173]. In CD8^+^ T cells, Rasal1 binding to ZAP-70 directly inhibits ZAP-70 activation. Together with its RasGAP activity, it therefore contributes to negatively regulate CD8^+^ T cell activation and anti-tumor immunity [98]. Furthermore, cancer cells can operate inhibitory mechanisms other than those induced by inhibitory surface molecules to escape immune surveillance. An example of this mechanism is represented by the intense late endosome/lysosome trafficking of melanoma cells at the lytic synapse, which promotes lysosome secretion and subsequent cathepsin-mediated Prf1 degradation as well as defective GzmB penetration into the target tumoral cell [174].

## 4. Conclusions and Perspectives

Over the past two decades, advances in our understanding of the TME have played a fundamental role in the development of new anti-cancer strategies designed to target the immune dysfunctions that are established in the TME. Cancer cells evolve to evade innate and adaptive cell-mediated tumor clearance by both secreting strategic soluble molecules and expressing surface inhibitory ligands. Among the Janus-faced components of the TME, CTLs are those with the most noticeable dual face. They are, on the one hand, intrinsically equipped with the most relevant and target-specific activities, but on the other hand, are most heavily affected by the suppressive activities of the TME. A more in-depth understanding of the molecular mechanisms controlling the normal killing activities of CTLs and how these mechanisms are made dysfunctional in the specific context of the immunosuppressive TME is expected to result in the development of strategies to deprive CTLs of their pro-tumoral functions, restoring them to their unambiguous identity of serial killers.

Traditional chemotherapy has major effects not only on cancer cells, but also on the TME, by strengthening the response of CTLs, increasing cancer antigenicity, and inhibiting immunosuppressive pathways [175]. More recently, therapies specifically targeting immune checkpoints (immune checkpoint inhibitors), especially the PD-1/PD-L1 axis, have led to remarkable advances in treating several malignancies. However, most patients do not respond to immune checkpoint inhibitors and even develop resistance, spurring the search for new therapeutic TME targets. The molecular mechanisms whose dysfunction in CTL is related to cancer progression are only beginning to be elucidated. The expanding evidence of the immunomodulatory function of the TME underscores the need to study these mechanisms in the context of the TME. Although this is a daunting task due to the multitude of cell types that form the TME and the complexity of their interaction network, unravelling the interplay of CTLs with the TME will bring us a major step forward to the identification of new therapeutic targets for cancer treatment based on counteracting CTL suppression by the TME. Still, the knowledge accumulated to date has already led to a revolution in cancer treatment, as witnessed by the current checkpoint inhibitor-targeted therapies. Combination therapies targeting two different immune checkpoints are one of the most promising approaches, as exemplified by the interplay of LAG-3 with PD-1 [176]. PD-1 and LAG-3 are extensively co-expressed in CD8^+^ TILs [177] and cooperate to suppress their cytotoxic effector functions [178]. Currently, clinical trials are ongoing to explore the therapeutic benefits of simultaneously targeting LAG-3 and PD-1 [179]. Intriguing co-regulatory mechanisms have also been reported for LAG-3 and CTLA-4, which were found to be co-expressed in CD8^+^Foxp3^+^ Tregs, where they participate in immune tolerance through a co-inhibitory signaling pathway, leading to the suppression of alloreactive T cell responses [176,180]. Interestingly, the anti-CTLA-4 antibody, ipilimumab, increases the frequencies of LAG-3-expressing TILs in metastatic melanoma patients [181], further supporting co-regulatory mechanisms for these receptors.

TIM-3 and PD-1 are also significantly co-expressed in cancer. In preclinical tumor models, both cytotoxic functions and expression of the effector cytokines IL-2, IFN-γ and TNF-α are severely suppressed in CD8^+^ T cells co-expressing TIM-3 and PD-1 [182]. Interestingly, PD-1-directed pharmacological therapies result in TIM-3 upregulation [183], and combined anti-PD-1 and anti-TIM-3 therapies in mouse models of cancer result in the substantial recovery of T cell responses, compared to single agents [184], supporting the notion that combined therapies against these surface molecules might be of greater impact than single-agent therapy in cancer treatment.

A new phase I/II clinical trial (NCT03459222) was recently opened to investigate the efficacy of co-targeting LAG-3, PD-1, and CTLA-4, which, by further enhancing the efficacy of single and double targeting approaches, might become a novel combinatorial strategy for cancer treatment in the near future. In this context, it is noteworthy that PD-1, LAG-3, and TIM-3 expression is coordinately increased in gastric cancer patients with better disease prognosis [185].

Ligands of checkpoint inhibitors are also promising targets for anti-cancer therapy. PD-L1 blockade by specific antibodies, which are currently approved treatment options for a broad range of cancer types [186], has noticeable effects on the CTL compartment, with enhanced tumor-specific cytotoxic activity and release of GzmB, Prf, and IFN-γ at the tumor site [187]. A report recently published by Yang and colleagues suggests the TIM-3 ligand Galectin-9 (Gal-9) as a target for immunotherapy based on the fact that (i) high Gal-9 expression correlates with poor prognosis in multiple human cancers [188], and (ii) in PD-1^+^TIM-3^+^ T cells, PD-1 sequesters Gal-9, hampering its binding to TIM-3 with subsequent TIM-3-dependent T cell death, thereby contributing to the persistence of the exhausted T cell population [189].

The building and persistence of a pro-tumoral microenvironment requires angiogenesis, a process which, by bringing new vessel branches to the tumor site, promotes tumor growth, local invasion, and metastasis. Once again, TME components strongly contribute to angiogenesis mainly generating hypoxia, a state of low oxygen tension common in cancer, which associates to abnormal vasculature and ultimately promotes tumor invasiveness and metastasis [190]. Of note, in the hypoxic TME, activated signaling molecules, such as hypoxia-inducible factor 1 (HIF-1) inhibit both innate and adaptive immune components by inducing the expression of immunosuppressive factors and immune checkpoint molecules, including VEGF and PD-1/PD-L1 [191], highlighting a network connecting angiogenesis, hypoxia, and immune system suppression. Angiogenesis is a recognized hallmark of cancer, often associated with increased aggressiveness and poorer prognosis [191,192]. Agents designed to specifically target VEGF and/or its cognate receptor VEGFR could be, therefore, considered promising candidates to block angiogenesis and ameliorate cancer prognosis [193].

In this multifaceted scenario, where each molecule within the TME can potentially become an interesting new target for cancer treatment, therapies combining anti-angiogenic drugs to immune checkpoint inhibitors represent the new horizon to explore. These new pharmacological candidates, some of which are in phase III clinical trials [194], could improve tumor outcomes by overcoming resistance to cancer immunotherapy via tumor vessel normalization.

## Figures and Tables

**Figure 1 ijms-22-11221-f001:**
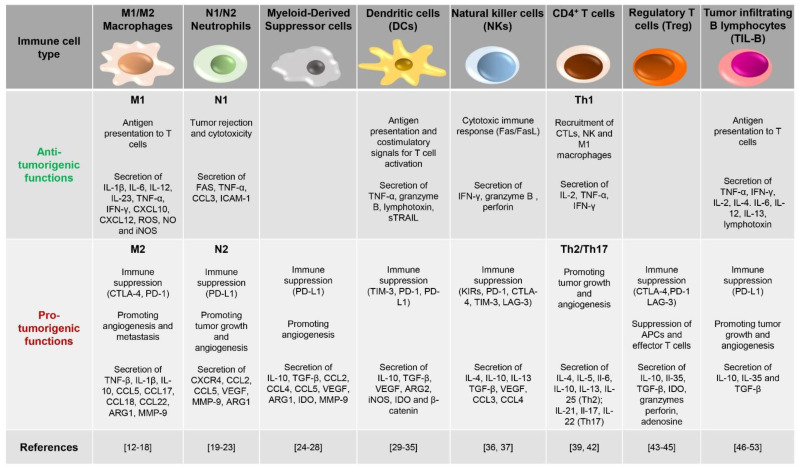
Schematic representation of key immune cells found in the tumor microenvironment. Depending on the tumoral context, immune cells exert either anti-tumorigenic or pro-tumorigenic functions. The balance between these two opposite functions determines the outcome of the immune response against cancer.

**Figure 2 ijms-22-11221-f002:**
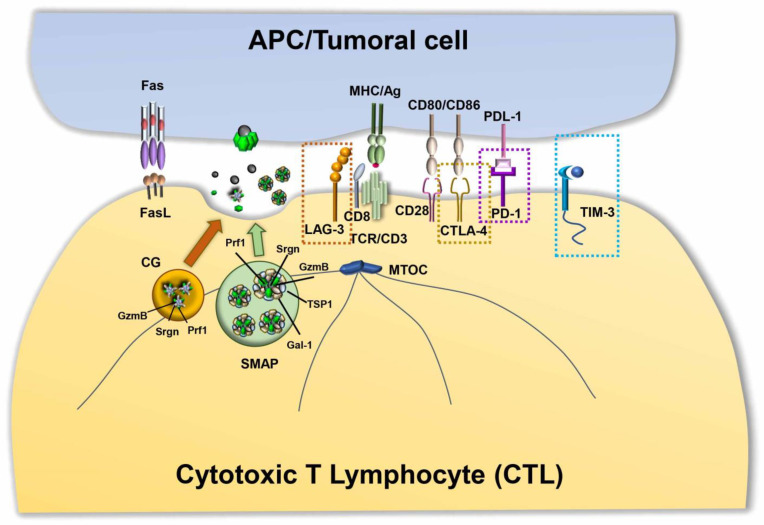
The cytotoxic immune synapse. Upon TCR recognition of tumoral antigens, CD8^+^ T cells activate cytotoxic mechanisms to kill target cells. The two classical cytotoxic mechanisms are represented by the release of cytotoxic granules (CGs), with perforin and granzymes released into the synaptic cleft, and the Fas/FasL pathway. Together with these well-characterized mechanisms of cytotoxicity, a new mechanism based on SMAP release contributes to tumoral cell killing. Signals triggered by the inhibitory receptors CTLA-4, PD-1, LAG-3 and TIM-3 antagonize TCR-dependent signaling, causing abnormalities in IS assembly and dysfunctions in the lytic granule transport and release.

**Figure 3 ijms-22-11221-f003:**
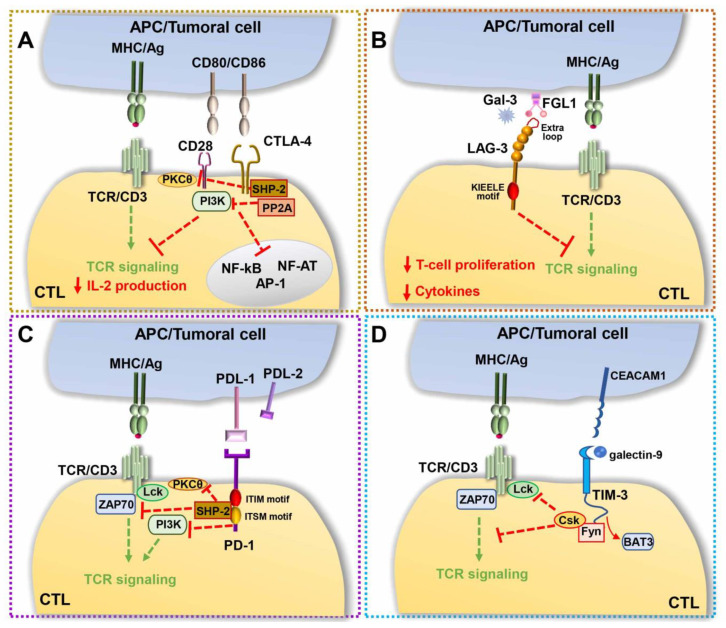
Immune checkpoints. Inhibitory signals triggered by CTLA-4/CD80 (CD86) (**A**), LAG-3/MHCll (or Gal-3 and FGL1) (**B**), PD-1/PD-L1(**C**), and TIM-3/Gal-9 (**D**) axes suppress the TCR-dependent signaling pathways, thereby contributing to the balance between self-tolerance and tumor cell clearing. Gal-3, Galectin-3; FGL1, fibrinogen-like protein 1; Gal-9, Galectin-9.

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
