# Peer review of "Nature vs. Nurture: The Two Opposing Behaviors of Cytotoxic T Lymphocytes in the Tumor Microenvironment"

_ijms, 2021, doi:10.3390/ijms222011221_

Round 1
Reviewer 1 Report
The review by Capitani et al. describe the opposing activities of immune cells populating the tumor microenvironment. The review is well-written. It illustrates both the pro-tumoral and the anti-tumoral roles played by most immune subsets in the TME and it focuses particularly on the co-inhibitory receptors on CD8 T cells. I have only two remarks:
-) metabolic alteration within the TME can have a profound effect on immune cell functionality. Several metabolites produced by cancer cells can impair immune response. Maybe these aspects could be included shortly in the manuscript to give a more comprehensive view of the immune cell alterations within the TME (also because several co-inhibitory receptors affect cell metabolism).
-) Overall, it seems that several cytokines can induce several immune cell types towards a pro-tumoral phenotype, while others promote an anti-tumoral one. Could it be useful, at discretion of the authors, to add a small paragraph to recapitulate how different cytokines may play a role in this dual game?
Author Response
Reviewer #1
Point 1. "Metabolic alteration within the TME can have a profound effect on immune cell functionality. Several metabolites produced by cancer cells can impair immune response. Maybe these aspects could be included shortly in the manuscript to give a more comprehensive view of the immune cell alterations within the TME (also because several co-inhibitory receptors affect cell metabolism)"
As requested, short paragraphs describing metabolic alterations observed in the TME and their crosstalk with co-inhibitory receptors have been included in the Introduction section and in section 2.2
Point 2. " Overall, it seems that several cytokines can induce several immune cell types towards a pro-tumoral phenotype, while others promote an anti-tumoral one. Could it be useful, at discretion of the authors, to add a small paragraph to recapitulate how different cytokines may play a role in this dual game?".
We agree with the Reviewer that the dual role of cytokines in the TME has acquired increasing importance in cancer pathology in the last decade. We have mentioned some of the cytokines implicated in immune cell functions both in the text (section 1) and in Figure 1, supported by bibliography. However, this is a broad field that is difficult to summarize in a short paragraph and additionally does not represent the focus of our review. We therefore only mentioned the relevance of cytokines in promoting the generation of pro-tumoral phenotypes in immune cells in the Introduction, referring to an exhaustive review on this topic. Relevant cytokines are then individually mentioned in the related parts of the manuscript.
Reviewer 2 Report
Nagaja Capitani et al uncovered key elements regarding two opposing behaviors of cytotoxic T lymphocytes in the tumor microenvironment.
Points to be addressed:
1) The rationale of why the authors came up with this review.
2) What is the information that is not exactly available that motivated the authors to come up with this information. What are the current caveats and how do the authors highlight the current research in answering them? If not they need to address in future directions.
3) In Section 3 Adaptive immune cells The authors correctly state that furthermore, TIL-B cells produce lymphotoxin and VEGF at the tumor site, which promote angiogenesis and support tumor progression. This reviewer personally misses some recent evidences regarding novel aspects of some cancer subtype. Ref 49 appropriately addresses prostate cancer (PCa). In this regard, novel evidences bear a clinical perspective for high-risk PCa patients with low miR-221-3p levels since this could predict a favourable TKI response. Apart from this therapeutic niche, a partially oncogenic function of miR-221-3p as an escape mechanism from VEGFR2 inhibition has been identified. tumors grow and evolve through a constant crosstalk with the surrounding microenvironment, and emerging evidence indicates that angiogenesis and immunosuppression frequently occur simultaneously in response to this crosstalk. Accordingly, strategies combining anti-angiogenic therapy and immunotherapy seem to have the potential to tip the balance of the tumor microenvironment and improve treatment response (please refer to PMID: 32131507).
4) Does this role of endothelial cells in angiogenesis in a tumor micro-environment involve hypoxia? Since hypoxia is a key factor for angiogenesis, the authors need to substantiate.
5) The authors need to highlight what new information the review is providing to enhance the research in progress.
Author Response
Reviewer #2
Point 1. " The rationale of why the authors came up with this review ".
According to the Reviewers’ request, we have pointed out the rationale of this review in the last paragraph of the Introduction section.
Point 2. " What is the information that is not exactly available that motivated the authors to come up with this information. What are the current caveats and how do the authors highlight the current research in answering them? If not they need to address in future directions."
We have better delineated the rationale of this review as detailed in the response to point 1 and discussed how current research is addressing open issues in the section "Conclusions and future directions".
Point 3. "In Section 3 Adaptive immune cells The authors correctly state that furthermore, TIL-B cells produce lymphotoxin and VEGF at the tumor site, which promote angiogenesis and support tumor progression. This reviewer personally misses some recent evidences regarding novel aspects of some cancer subtype. Ref 49 appropriately addresses prostate cancer (PCa). In this regard, novel evidences bear a clinical perspective for high-risk PCa patients with low miR-221-3p levels since this could predict a favourable TKI response. Apart from this therapeutic niche, a partially oncogenic function of miR-221-3p as an escape mechanism from VEGFR2 inhibition has been identified. tumors grow and evolve through a constant crosstalk with the surrounding microenvironment, and emerging evidence indicates that angiogenesis and immunosuppression frequently occur simultaneously in response to this crosstalk. Accordingly, strategies combining anti-angiogenic therapy and immunotherapy seem to have the potential to tip the balance of the tumor microenvironment and improve treatment response (please refer to PMID: 32131507)."
In this point the Reviewer brings attention to a very relevant issue. We discussed this issue in the last paragraph of the “Conclusions and perspectives” section, where we have highlighted the relevance of angiogenesis in tumor progress and underlined combined immune- and anti-angiogenic therapies as potential new approaches to improve anti-cancer responses.
Point 4. "Does this role of endothelial cells in angiogenesis in a tumor micro-environment involve hypoxia? Since hypoxia is a key factor for angiogenesis, the authors need to substantiate.".
We have addressed this point together with point 3 in the last paragraph of the “Conclusion and perspectives” section, where we introduced hypoxia as one of the main causes of neo-angiogenesis in cancer.
Point 5. " The authors need to highlight what new information the review is providing to enhance the research in progress."
We have highlighted the new aspects reviewed in this manuscript that can contribute to improve the research in this field in the first and second paragraphs of the “Conclusion and perspectives” section.
Round 2
Reviewer 2 Report
The authors have clarified several of the questions I raised in my previous review. Most of the major problems have been addressed by this revision.